# Mathematical Modeling, Analysis and Evaluation of the Complexity of Flight Paths of Groups of Unmanned Aerial Vehicles in Aviation and Transport Systems

**Andrey Kositzyn [1], Denis Serdechnyy [2], Sergey Korchagin [3], Ekaterina Pleshakova [3,\*], Petr Nikitin [3] and Natalia Kurileva [4]**

1   Russian Academy of Sciences, Profsoyuznaya, 65, 117806 Moscow, Russia; Kositzyn@gmail.com
2   Department of Innovation Management, State University of Management, Ryazansky Pr., 99, 109542 Moscow, Russia; DV_Serdechnyj@guu.ru
3   Department of Data Analysis and Machine Learning, Financial University under the Government of the Russian Federation, Shcherbakovskaya, 38, 105187 Moscow, Russia; SAKorchagin@fa.ru (S.K.); pvnikitin@fa.ru (P.N.)
4   Department of General and Professional Education, Mari State University, Lenin Square 1, 424000 Yoshkar-Ola, Russia; knlmspi@rambler.ru
\*   Correspondence: pleshakova_es@mail.ru

**Abstract:** Recently, we have seen the rapidly growing popularity of unmanned aerial vehicles. This is due to some advantages, namely portability, the ability to fly over hard-to-reach areas without human intervention. They are also widely used for commercial purposes, agriculture, delivery, automation in warehouses. The potential of unmanned aerial vehicles is vast and demonstrates promising opportunities. However, when using these devices, the issue of safety is acute. This article presents a developed software application that is used to improve the efficiency of flight research of groups of unmanned aerial vehicles, based on a new method for assessing flight safety by comparing the complexity of specified air routes. A practical approach to modeling and evaluating the search for a safe way is proposed. A suitable method of research is computer and simulation modeling. It is suggested to use the spectrum of dynamic characteristics of the sequence as a formal attribute for analyzing routes. The method is illustrated by an example of comparing air trajectories according to the flight safety criterion. The software application is intended for use in the educational process when training specialists in transport security, robotics, and system analysis.

**Keywords:** software application; groups of unmanned aerial vehicles; coding; mathematical modeling; flight trajectories; trajectory planning





## 1. Introduction

Over the past few years, unmanned aerial vehicles have gained enormous popularity [1,2]. They are used in various industries: sensing, target observation, scientific data collection, disaster management, smart cities, search and rescue operations, food delivery, etc. Unmanned aerial vehicles have some advantages, particularly portability, efficiency, and low cost, to facilitate unmanned aerial cars [3]. However, along with the benefits of unmanned aerial vehicles, some disadvantages hinder their use. Safety in this area is particularly acute, mainly related to ensuring the safe operation of unmanned aerial vehicles during the flight. The safe operation of unmanned aerial vehicles is the most critical and urgent task since any unexpected deviation from the specified route will entail risks.

The main problem when conducting a full-scale experiment is its high cost [4–7]. In this case, computer and simulation modeling is an effective and sometimes possible method of research and training [8–10]. To solve this problem, the article presents a developed software application that allows you to compare the complexity of given routes, which is based on splitting ways into parts and analyzing the properties of each of the

sections, taking into account their location on the way. Thus, on the one hand, route planning can be considered a multi-purpose optimization task. However, on the other hand, the trajectory planning task is more complex due to many constraints, for example, complex environmental constraints, physical constraints, etc. [11]. Therefore, it is a difficult optimization task with multitasking and multitasking rules in its essence [12].

Currently, there are three classes of approaches to solving the problem of route planning. The first class is based on graph algorithms [13–17]. The second class is based on a typical heuristic method that generates a trajectory with the lowest cost guide from a given initial node to the target node [18]. As a result, the estimated volume of searching for the optimal route will increase [19–21]. The third class is based on the algorithms of evolutionary calculations. Several ways are initialized randomly, and the specified evaluation criteria calculate the objective function's corresponding values. These approaches have satisfactory performance, are more flexible and efficient. However, despite this, they have several disadvantages, such as local optima and a slow global convergence rate. The general scheme of planning the route of unmanned aerial vehicles, based on the algorithms of evolutionary calculations, is shown in Figure 1.

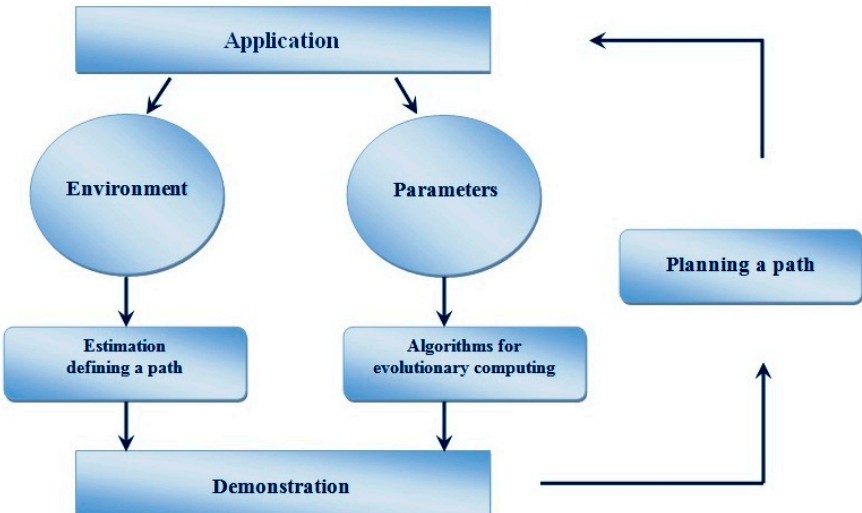

**Figure 1.** General planning system based on evolutionary calculation algorithms.

Mathematical modeling of the flight of a group of unmanned aerial vehicles is a key issue in the study of flight safety. A mathematical model for assessing the complexity and safety of the flight path of a group of unmanned aerial vehicles is described, which has shown good results, which allowed us to develop software based on the results of the study that allows us to calculate the complexity and safety of the flight paths of groups of unmanned aerial vehicles, download maps, etc.

This article implements an effective method for estimating the complexity of unmanned aerial vehicles' flight paths using a formal apparatus. For the analysis of routes, it is proposed to use the spectrum of dynamic characteristics of the sequence.

In our work, we propose to use a range of dynamic characteristics to analyze and build an estimate of the complexity of a route represented as a set of sequences of code signs. In this spectrum, dynamic parameters representing the rules for constructing a sequence using recurrent forms of various orders are systematized.

The values of the indicators for each of the properties that characterize the constructed sections are encoded with the symbols of the finite alphabet. When they are ordered according to the chosen direction of traversing the areas, a sequence is formed that describes the dynamics of changes in the corresponding property when moving along the route. The set created by the lines constructed for all the properties under consideration is a complete description of the way since it contains information about the value of the indicator of each of the properties for any section of the way. Thus, the software application

allows students to conduct simulation modeling and analyze the complexity of routes based on data obtained during field experiments and synthetic data obtained during a computational investigation.

When solving the problem, mathematical models known in the theory of regression analysis are used. The mathematical model of air transport was taken as a basis, which is used in many studies, for example, when planning ground operations, as well as when controlling flight processes.

It is proposed to use the spectrum of dynamic characteristics as a formal apparatus for analyzing and constructing an estimate of the complexity of a route represented as a set of sequences of code signs [22–28]. In this spectrum, dynamic parameters representing the rules for constructing a line using recurrent forms of various orders are systematized [29] (Figure 2).

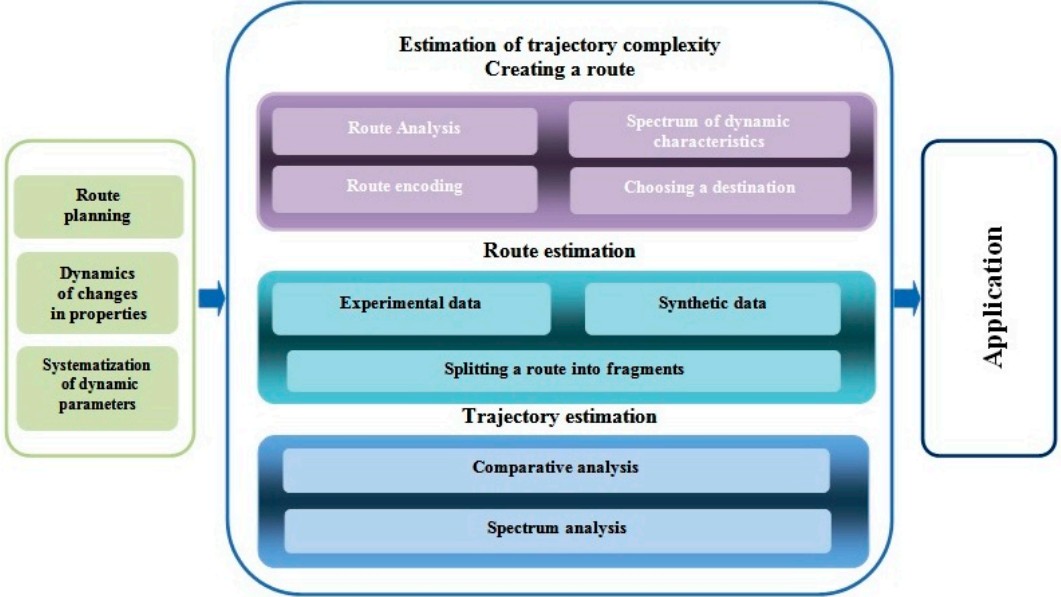

**Figure 2.** Overview of the proposed algorithm for constructing the optimal path.

## 2. Models and Methods

Let $U = \{u_1, u_2, \ldots, u_k\}$ be a finite set and $\xi \in U^*$. The spectrum $\Omega(\xi)$ of the dynamic characteristics of the sequence $\xi$ has a hierarchical structure consisting of five levels $\Omega(\xi) = (\Omega_0(\xi), \Omega_1(\xi), \Omega_2(\xi), \Omega_3(\xi), \Omega_4(\xi))$. For the formal definition of these levels, we will introduce the following notation.

For any sequence $\overline{\xi} \in U^\gamma$, the most minor order of the recurrent form defining the sequence $\overline{\xi}$ will be denoted by $m_0(\overline{\xi})$. For any sequence $\overline{\xi} \in U^\gamma$ and $m \in N^+$, where $1 \leq m \leq m_0(\overline{\xi})$, the most extensive length of the initial segment of the sequence $\overline{\xi}$, represented by the recurrent form of order m, will be denoted by $d^m(\overline{\xi})$. For any sequence $\overline{\xi} \in U^\gamma$ and $m \in N^+$ the number of shifts of recurrent forms of order m required in determining the sequence $\overline{\xi}$, we represent $r^m(\overline{\xi})$. For any sequence $\overline{\xi} \in U^\gamma$ and $m \in N^+$, where $1 \leq m \leq m_0(\overline{\xi})$ and j, where $1 \leq j \leq r^m(\overline{\xi})$, the length of the j segment in the definition of the sequence $\overline{\xi}$ will be represented by $d_j^m(\overline{\xi})$ [1].

Using the notation we have entered, we define the spectrum of parameters that characterize the sequence as the following structure:

$$\Omega_0(\overline{\xi}) = \langle m_0(\overline{\xi}) \rangle; \tag{1}$$

$$\Omega_1(\overline{\xi}) = \langle d^1(\overline{\xi}), d^2(\overline{\xi}), \ldots, d^\alpha(\overline{\xi}) \rangle; \tag{2}$$

$$\Omega_2(\overline{\xi}) = \langle r^1(\overline{\xi}), r^2(\overline{\xi}), \ldots, r^\alpha(\overline{\xi}) \rangle; \tag{3}$$

$$\Omega_3(\overline{\xi}) = \left\langle \Omega_3^1(\overline{\xi}), \Omega_3^2(\overline{\xi}), \ldots, \Omega_3^\alpha(\overline{\xi}) \right\rangle; \tag{4}$$

where $\alpha = m_0(\overline{\xi})$ and $\Omega_3^j(\overline{\xi}) = \left\langle d_1^j(\overline{\xi}), d_2^j(\overline{\xi}), \ldots, d_{n_j}^j(\overline{\xi}) \right\rangle$, $n_j$ is the number of the last segment in the definition of the sequence $\overline{\xi}$ as a sequence of features defined by separate recurrent forms of order j.

$\Omega_4(\overline{\xi}) = \Theta(\Omega_3(\overline{\xi}))$, where $\Theta$ is the operator for replacing in $\Omega_3(\overline{\xi})$ the values of the lengths of the segments with the weights of the recurrent forms used to determine the features.

The fifth level $\Omega_4(\overline{\xi})$ of the spectrum $\Omega(\overline{\xi})$ adds an estimate of the complexity of the rules and the use cases of the rules to the characterization of sequence $\overline{\xi}$ in terms of the number of rule changes that determine the relative positions of elements in the line and the values of the rule scopes represented at levels $\Omega_1(\overline{\xi})$–$\Omega_3(\overline{\xi})$. Then, in a reasonably general case, you can enter the weights of the rules (recurrent forms) and the consequences of implementing the regulations. For example, for each step of applying recurrent conditions of $F(z_1^0, z_2^0, \ldots, z_m^0) = z_{m+1}^0$, for a set, set weight in numerical form, and the sum of all the stages of using the recursive structure for the sequence relies on weight forms.

## 3. Route Construction Algorithm

Let us set the flight route of a group of unmanned aerial vehicles, represented by a curved line on the map. The mathematical apparatus of the considered spectra is designed to systematize the numerical characteristics of the sequence. Therefore, in the described method of assessing the complexity, information about the route is converted into a route code—a set of lines of code signs.

The main hypothesis is that the absolute value of the height difference between the end and start points of any section cannot exceed twice the height difference between the level lines. The routes of unmanned aerial vehicles, in accordance with the task statement, are divided into stages. The values of the property $R_1$ and $R_2$ indicators for the sections of each route are calculated using the map. The obtained values are encoded with the symbols of the alphabets A1 and A2.

The developed mathematical models are based on well-known approaches to modeling air transportation, published in authoritative and peer-reviewed publications. The results obtained using the proposed new methods and approaches are compared with the results of the application of known methods. The limits of applicability of the proposed methods and approaches are discussed. All the assumptions made during the work are described in detail, with a description of the limits of applicability and the impact on the final result. The reproducibility of the obtained results is ensured by a detailed description of all stages of data processing, indicating the parameters of the processing methods and the software used. The results of theoretical analysis and numerical modeling are confirmed by the results obtained during experimental studies.

The first stage of constructing a route code represents the route as a finite number of disjoint sections. It is proposed to select such areas of the way by dividing the curve line describing the road on the map into fragments and comparing the sizes of the entire route to the constructed elements. This partitioning can be implemented in various ways, for example:

- Splitting into fragments of the same length.
- Splitting into fragments corresponding to the selected typical sections of the route.
- Splitting into fragments, selected by the properties of the airspace through which the route is laid.

In this article, the method was illustrated by dividing the route under study into fragments of the same length.

The constructed fragments are segments of an abstract curved line representing the plane's actual curve projection. To assess the complexity of the flight along the route, it is necessary to interpret these fragments, comparing them with sections of the way according to the scale and location of the level lines on the map, and highlight the properties that are essential for assessing the complexity of the flight for each of the sections [30–33] (Figure 3).

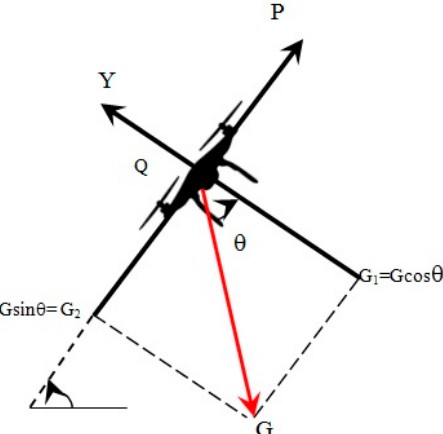

**Figure 3.** Schematic representation of an unmanned aerial vehicle.

As such features, the following can be used:

- The angle of climb or descent in the direction of movement along the route. The value of this property is the value of the angle between the perpendicular to the surface along which the route passes and the tangent to the real curve of the route line. The perpendicular and tangent are constructed from the same point selected on the plot.
- The average angle of climb or descent in the direction of movement along the route. The value of this property is obtained by calculating the ratio of the integral value along the curve of the line corresponding to this section from the value of the previous property at each point of the area to the length of this section.
- The angle of the transverse slope. The value of this property is the value of the angle between the vertical and the line, the direction of which is set by the vector product of the direction of movement along the route and the external normal to the surface, constructed from the same point selected on the section.
- The average angle of the transverse slope.
- Properties characterize the specifics of the space in which the route section is located: the presence or absence of restricted zones, their shape, size, atmospheric phenomena that affect traffic on this section, etc.
- Properties that characterize the specifics of the trajectory of this section of the route: the number of turns, the radius and angle of rotation, the distance between turns, etc.

In this article, we will consider the properties that characterize the difference in the heights of the points located at the ends of the plot and the angle of the transverse slope. As the point at which the lines involved in calculating the values of the indicator of the last property were plotted, the point corresponding to the middle of the fragment associated with the route section was selected (Figure 4).

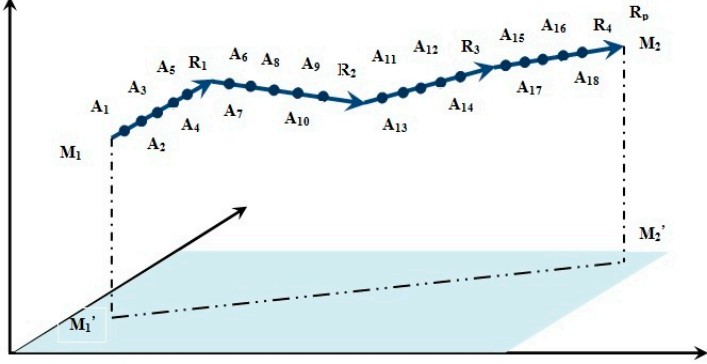

**Figure 4.** Flight path.

We denote the properties of $R_1$, $R_2$,..., $R_p$ under consideration. For each property $R_i$, where $1 \leq i \leq p$, a set $A_i$ is defined, the elements of which will be used to encode the values of the property indicator.

Let us compare the numbers of the constructed sections of the route according to the order in which the sections meet when traversing the way in the specified direction. Let these numbers be $1, 2, \ldots, q$, then the value of the indicator of the i property, where $1 \leq i \leq p$, for the j section of the route, where $1 \leq j \leq q$, will be denoted as $r_{ij}$. For each fixed i, where $1 \leq i \leq p$, the $a_{ij}$ codes of the values of $r_{ij}$, where $1 \leq j \leq q$, make up the sequence $\alpha_i = \langle a_{i1}, a_{i2}, \ldots, a_{iq} \rangle$. The resulting set of $\alpha_1, \alpha_2, \ldots, \alpha_p$ lines is the code of the route under study.

Aircraft routes, which are divided into stages. We apply the spectrum of dynamic characteristics of the sequence to compare the complexity of the two stages of the aircraft route formed by the initial and final fragments of the way. We will consider these stages as two different routes, for which we will introduce the notation (corresponds to the initial stage) $M_1$ and $M_2$ (corresponds to the final stage).

Next, the code of each of the routes $M_1$ and $M_2$ is built. To divide the ways into sections, the value of the length of the fragments was chosen equal to 80 km in real terms. This choice allowed to map these fragments to areas of the route that intersect no more than one level line on the map. Finally, the constructed sections are characterized by the properties $R_1$ (height difference) and $R_2$ (angle of descent or climb), for the values of the indicators necessary to determine the alphabets of code characters.

Due to the choice of the length of the fragments, it can be argued that the absolute value of the height difference between the end and start points of any section cannot exceed twice the height difference between the level lines. Therefore, as the alphabet $A_1$, we take the set of integers lying in the segment $[-20, 20]$: $A_1 = \{ a_1 \in Z \mid -20 \leq a_1 \leq 20 \}$. Let us put $A_2 = \{ a_2 \in Z \mid -45 \leq a_2 \leq 45 \}$, since the maximum allowable role of unmanned aircraft is $45°$.

The map calculates the values of the $R_1$ and $R_2$ property indicators for the sections of each route. The resulting values are encoded with the characters of the alphabets $A_1$ and $A_2$ by rounding the value of the property indicator to the nearest integer. These symbols form pairs of $\alpha_1^{(1)}, \alpha_2^{(1)}$ and $\alpha_1^{(2)}, \alpha_2^{(2)}$ sequences, which represent the route $M_1$ and $M_2$, respectively:

$$\alpha_1^{(1)} = <0, -1, 0, -3, 0, -1, -1, 0, -2, 0, -1, 0, -1, 0, -2, -2, -2, -1, -3, -2, -2, -1, -1, -1, -2, 0, -1, 2, 2, -2, -2, 0, \\ -1, 0, 0, -3, -4, -2, -4, -3, -3, 0, -1, 0, 1, 1, 0, 0, 0, -1 >; \tag{5}$$

$$\alpha_2^{(1)} = <0, -3, -2, -3, 1, 1, 1, 1, 1, 3, 2, 5, 5, 0, 5, 5, 0, 0, -5, -5, 0, 1, 2, 1, 1, 1, 1, 1, 0, -9, 0, 1, 0, 0, 0, 11, 1, 2, 8, -4, \\ 2, 2, 4, 4, 2, 1, 1, 4, 2, 0 >; \tag{6}$$

$$\alpha_1^{(2)} = <-6, -8, -9, -7, -8, -5, -5, -10, -8, -6, -4, -3, -8, -8, -6, -8, -8, -8, -9, -10, -10, 0, 16, 16, 13, 8, 9, -20, \\ -29, -20, -16, -20, -16, -16, -10, -8, -10, 0, 0, 0, 9, 6, 9, 13, 6, 6, 4, 3, 3, 0 >; \tag{7}$$

$$\alpha_2^{(2)} = <0, 0, 0, 0, 16, 16, 13, 0, 0, 0, 0, -16, 0, 0, -16, 13, 16, 29, 13, 29, 13, 9, 9, 8, 4, 6, 10, 10, 13, 29, 0, 0, 0, 0, 0, \\ 0, 13, 9, 9, 16, -16, -8, -4, -6, -10, -8, -3, -8, -13, -13 > . \tag{8}$$

The objective function is to maximize the total reward for UAV exploration, where time and resources are subject to many restrictions. Equation (3) is an objective function that indicates that each target area is maximized by multiplying the reward for exploration after the exploration time by the value of the target area. Where $x_{ijk}$ is a decision variable, and ci is a fixed coefficient that measures the strategic value of each target area. $R(tj)$ is the reward for exploration of each target area, determined by the decision variable tt. Equation (4) indicates that the total flight time of each UAV to complete missions is less than the maximum flight time of the UAV. Where $t_{ijk}$ is the flight time required for the UAV k to fly from the target area i to the target area j, $t_j$ is the reconnaissance time allocated by

the UAV k to the target area j, as well as the flight time of the UAV k flying to the target area j. $T_f$ is the maximum flight time of the UAV k. Equation (5) indicates that the total exploration time of each UAV is less than the maximum opening time of the reconnaissance sensor. Where Tr is the maximum opening time of the reconnaissance sensor carried by the UAV. Equations (6) and (7) guarantee that each target area can be explored by one and only one UAV [34].

The comparison of routes $M_1$ and $M_2$ in terms of complexity is proposed to be carried out by analyzing the spectra of dynamic characteristics constructed for the sequences $\alpha_1^{(1)}$ and $\alpha_1^{(2)}$, $\alpha_2^{(1)}$ and $\alpha_2^{(2)}$. The numerical values of the dynamic parameters of the $\Omega\left(\alpha_1^{(1)}\right)$ and $\Omega\left(\alpha_1^{(2)}\right)$ ranges are presented below:

$$\Omega_0\left(\alpha_1^{(1)}\right) = 5; \tag{9}$$

$$\Omega_1\left(\alpha_1^{(1)}\right) = \langle 3, 6, 12, 34, 50 \rangle; \tag{10}$$

$$\Omega_2\left(\alpha_1^{(1)}\right) = \langle 19, 9, 5, 1, 0 \rangle; \tag{11}$$

$$\Omega_3\left(\alpha_1^{(1)}\right) = \left\langle \Omega_3^1\left(\alpha_1^{(1)}\right), \Omega_3^2\left(\alpha_1^{(1)}\right), \ldots, \Omega_3^5\left(\alpha_1^{(1)}\right) \right\rangle; \tag{12}$$

$$\Omega_3^1\left(\alpha_1^{(1)}\right) = \langle 3, 3, 3, 4, 5, 4, 4, 2, 4, 4, 3, 3, 4, 2, 5, 3, 4, 3, 4, 2 \rangle; \tag{13}$$

$$\Omega_3^2\left(\alpha_1^{(1)}\right) = \langle 6, 7, 5, 5, 7, 4, 11, 13, 7, 3 \rangle; \tag{14}$$

$$\Omega_3^3\left(\alpha_1^{(1)}\right) = \langle 12, 5, 11, 14, 14, 9 \rangle; \tag{15}$$

$$\Omega_3^4\left(\alpha_1^{(1)}\right) = \langle 34, 20 \rangle; \tag{16}$$

$$\Omega_3^5\left(\alpha_1^{(1)}\right) = \langle 50 \rangle. \tag{17}$$

$$\Omega_0\left(\alpha_1^{(2)}\right) = 6; \tag{18}$$

$$\Omega_1\left(\alpha_1^{(2)}\right) = \langle 4, 9, 9, 9, 28, 50 \rangle; \tag{19}$$

$$\Omega_2\left(\alpha_1^{(2)}\right) = \langle 16, 6, 3, 3, 1, 0 \rangle; \tag{20}$$

$$\Omega_3\left(\alpha_1^{(2)}\right) = \left\langle \Omega_3^1\left(\alpha_1^{(2)}\right), \Omega_3^2\left(\alpha_1^{(2)}\right), \ldots, \Omega_3^6\left(\alpha_1^{(2)}\right) \right\rangle; \tag{21}$$

$$\Omega_3^1\left(\alpha_1^{(2)}\right) = \langle 4, 6, 5, 3, 2, 3, 3, 5, 5, 4, 3, 3, 7, 2, 3, 4, 4 \rangle; \tag{22}$$

$$\Omega_3^2\left(\alpha_1^{(2)}\right) = \langle 9, 10, 13, 7, 4, 16, 3 \rangle; \tag{23}$$

$$\Omega_3^3\left(\alpha_1^{(2)}\right) = \langle 9, 11, 14, 25 \rangle; \tag{24}$$

$$\Omega_3^4\left(\alpha_1^{(2)}\right) = \langle 9, 22, 5, 26 \rangle; \tag{25}$$

$$\Omega_3^5\left(\alpha_1^{(2)}\right) = \langle 28, 27 \rangle; \tag{26}$$

$$\Omega_3^6\left(\alpha_1^{(2)}\right) = \langle 50 \rangle. \tag{27}$$

A comparative analysis of the levels $\Omega_3\left(\alpha_1^{(1)}\right)$ and $\Omega_3\left(\alpha_1^{(2)}\right)$ shows that the lengths of the segments defined by the recurrent forms of the 1st and 2nd order in the sequences $\alpha_1^{(1)}$ and $\alpha_1^{(2)}$ have pretty close values. The large number (represented at levels $\Omega_2\left(\alpha_1^{(1)}\right)$ and $\Omega_2\left(\alpha_1^{(2)}\right)$) of changes in the conditions of these orders indicates that in the lines $\alpha_1^{(1)}$ and $\alpha_1^{(2)}$, at relatively small intervals, identical subsequences occur, followed by different characters.

The interpretation of sequences $\alpha_1^{(1)}$ and $\alpha_1^{(2)}$ components as routing $M_1$ and $M_2$, we can conclude that the motion along these routes meet close are the same (in the sense of the properties) of the set of regions such that the values of the properties characterizing the movement, during the group phase, differ. This difference complicates the work of the crew of the aircraft. Furthermore, determining the sequence $\alpha_1^{(1)}$ requires a more significant number of recurrent forms than the sequence $\alpha_1^{(2)}$, which allows us to characterize the route $M_1$ as more complex, and therefore more dangerous, for the crew than route $M_2$.

The following numerical values represent the spectra of the dynamic characteristics of sequences and:

$$\Omega_0\left(\alpha_2^{(1)}\right) = 3; \tag{28}$$

$$\Omega_1\left(\alpha_2^{(1)}\right) = \langle 5, 15, 50 \rangle; \tag{29}$$

$$\Omega_2\left(\alpha_2^{(1)}\right) = \langle 16, 4, 0 \rangle; \tag{30}$$

$$\Omega_3\left(\alpha_2^{(1)}\right) = \left\langle \Omega_3^1\left(\alpha_2^{(1)}\right), \Omega_3^2\left(\alpha_2^{(1)}\right), \Omega_3^3\left(\alpha_2^{(1)}\right) \right\rangle; \tag{31}$$

$$\Omega_3^1\left(\alpha_2^{(1)}\right) = \langle 5, 3, 7, 2, 3, 3, 4, 4, 7, 4, 2, 4, 4, 4, 4, 2 \rangle; \tag{32}$$

$$\Omega_3^2\left(\alpha_2^{(1)}\right) = \langle 15, 5, 17, 9, 12 \rangle; \tag{33}$$

$$\Omega_3^3\left(\alpha_2^{(1)}\right) = \langle 50 \rangle; \tag{34}$$

$$\Omega_0\left(\alpha_2^{(2)}\right) = 6; \tag{35}$$

$$\Omega_1\left(\alpha_2^{(2)}\right) = \langle 4, 4, 4, 11, 36, 50 \rangle; \tag{36}$$

$$\Omega_2\left(\alpha_2^{(2)}\right) = \langle 13, 6, 5, 3, 1, 0 \rangle; \tag{37}$$

$$\Omega_3\left(\alpha_2^{(2)}\right) = \left\langle \Omega_3^1\left(\alpha_2^{(2)}\right), \Omega_3^2\left(\alpha_2^{(2)}\right), \ldots, \Omega_3^6\left(\alpha_2^{(2)}\right) \right\rangle; \tag{38}$$

$$\Omega_3^1\left(\alpha_2^{(2)}\right) = \langle 4, 3, 6, 3, 2, 6, 3, 3, 6, 9, 4, 8, 3, 3 \rangle; \tag{39}$$

$$\Omega_3^2\left(\alpha_2^{(2)}\right) = \langle 4, 7, 4, 6, 8, 17, 16 \rangle; \tag{40}$$

$$\Omega_3^3\left(\alpha_2^{(2)}\right) = \langle 4, 9, 4, 7, 24, 17 \rangle; \tag{41}$$

$$\Omega_3^4\left(\alpha_2^{(2)}\right) = \langle 11, 27, 6, 18 \rangle; \tag{42}$$

$$\Omega_3^5\left(\alpha_2^{(2)}\right) = \langle 36, 19 \rangle; \tag{43}$$

$$\Omega_3^6\left(\alpha_2^{(2)}\right) = \langle 50 \rangle. \tag{44}$$

Analysis of the reduced $\Omega\left(\alpha_2^{(1)}\right)$ and $\Omega\left(\alpha_2^{(2)}\right)$ spectra shows that a recurrent order 6 is required to determine the sequence $\alpha_2^{(2)}$. In contrast, a recurrent form of order 3 is sufficient to decide on the line $\alpha_2^{(1)}$. This is because the line $\alpha_2^{(2)}$ contains several subsequences (up to 6 in length) consisting of a single code character. This symbol is 0; therefore, on route and $M_2$, in contrast to route $M_1$, there are segments of considerable length, on which there is no angle of transverse inclination. Consequently, the movement along these segments is less complicated than on the details with a transverse slope. Therefore, by property $R_2$, route $M_1$ is more complex and dangerous than route $M_2$.

Thus, based on the analysis of the spectra of the route codes $M_1$ and $M_2$ carried out using the Web application, it was concluded that the dynamics of the properties of $R_1$ and $R_2$ characterizes the route $M_1$ as more complex and dangerous than the route $M_2$.

## 4. Results

The article proposes a new method for assessing the complexity of the flight paths of a group of unmanned aerial vehicles using a formal route analysis apparatus used to determine the complexity and safety of the flight paths of groups of unmanned aerial vehicles. Based on the results of the research, a set of mathematical models and a prototype of software have been developed that allow assessing the complexity of the flight paths of groups of unmanned aerial vehicles and choosing the optimal route.

The application of the spectra described in mathematical models to assess the complexity of the route will be presented in the example of the path of unmanned aerial vehicles. For a visual representation of the work of mathematical models, a prototype of the software is provided (Figure 5). At the first stage, a map is loaded into the software application to simulate the flight paths of unmanned aerial vehicles. The prototype of the software can only work with maps on which the markup was previously carried out—the coordinates of the map areas. At this stage of research, we carry out the markup in manual mode. At the next stage, we create a data file in which we specify the coordinates of the start and endpoints of the flight of unmanned aerial vehicles, which is loaded into the program using the "upload data" button. This can be a file in the following formats .txt or .csv.

The "Plot route" button is used to calculate the safe trajectories of unmanned aerial vehicles. All calculations are based on mathematical models and algorithms described in Sections 2 and 3. There is also a function for analyzing the safety of the flight path, which is implemented when the "analysis" button is pressed. We have conditionally divided each route according to the level of flight safety (from a high level of danger to a low level of danger), the calculation of which is based on the mathematical models described in Section 2. The hazard criterion is the values of the spectra calculated using the mathematical model proposed by us. For greater clarity, in order not to give numerical values of the calculated spectra, we made a conditional color scale, where the high level of danger of trajectories is marked in red (corresponds to high values for the sum of properties P), the safest trajectories are marked in green (corresponds to low values for the sum of properties P) (Figure 6).

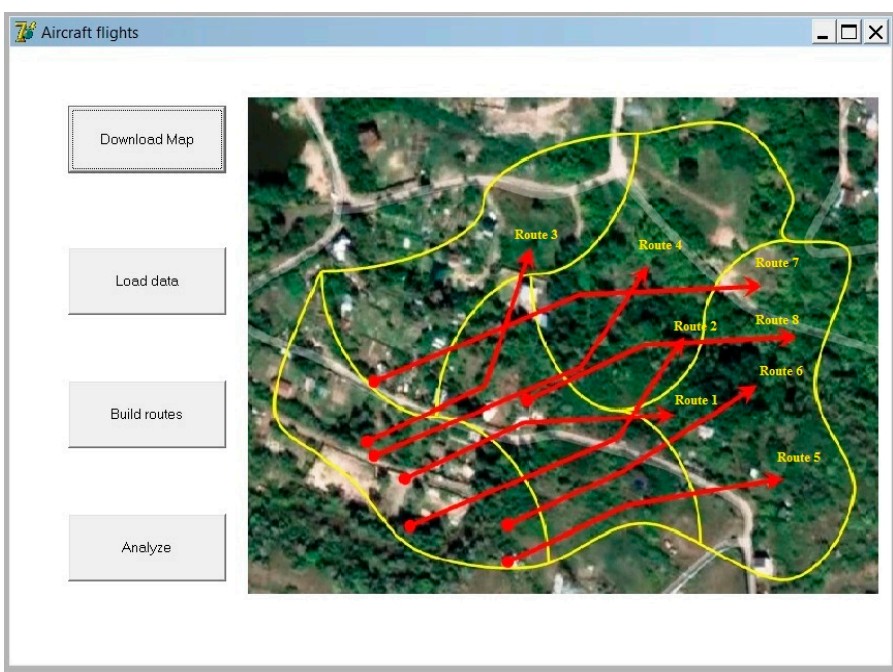

**Figure 5.** A fragment of the map of the flight area of unmanned aerial vehicles.

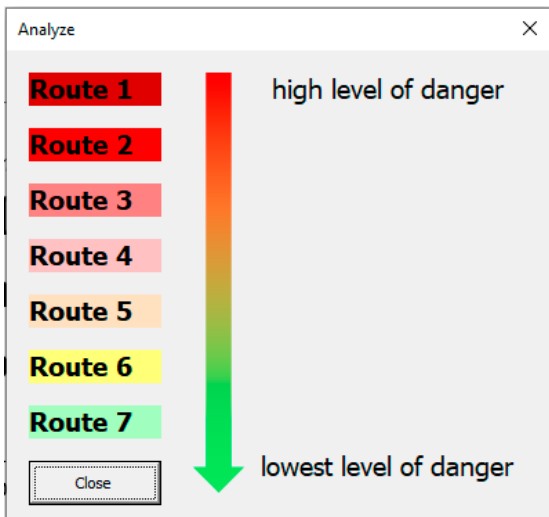

**Figure 6.** Functionality of the software (sections "Analysis").

As laboratory tests, we had data from a full-scale experiment with a small number of drones (we used 7 devices). We received information about the properties of the unmanned aerial vehicle (the properties are described in detail in Section 2), which were recorded on the memory card. Next, we worked with these data. In addition, to check the quality of our models and their compliance with experimental data, we used data on drone flights obtained in the sources [35–42].

At the moment, the prototype does not implement software synchronization with drones. However, we plan to do this using the Robot Operating System application development environment and the Gazebo software simulator, which allows you to import code written in C-like programming languages. They are used to test algorithms in a simulation environment and to disable as little equipment as possible. Both ROS and Gazebo work on Linux-we plan to use Ubuntu 16.04. A fragment of the flight area map was uploaded to the application (Figure 5). The software application is developed in the Delphi software environment.

At the moment, the prototype does not implement software synchronization with drones. However, we plan to do this using the Robot Operating System application development environment and the Gazebo software simulator, which allows you to import code written in C-like programming languages. They are used to test algorithms in a simulation environment and to disable as little equipment as possible. Both ROS and Gazebo work on Linux-we plan to use Ubuntu 16.04. A fragment of the flight area map was uploaded to the application. The software application is developed in the Delphi software environment.

In the future, it is planned to finalize the software package with the use of more drones and on more complex routes and the ability to synchronize the software with physical devices. The proposed models, algorithms, and software package can also be used for the safe planning of commercial air transportation. In the near future, tests of a group of unmanned aerial vehicles for agricultural purposes are planned.

The results of modeling and analysis of data on the flight paths of a group of unmanned aerial vehicles in aviation transport systems made it possible to determine based on the study of the spectra of route codes Route 1–7, conducted with the use of a software application that the dynamics of properties $R_1$ and $R_2$ characterizes the route Route 1 as a more dangerous and challenging route than Route 7.

At this stage, a prototype of the software product has been developed. In the future, it is planned to finalize the software package with the use of more drones and on more complex routes. The proposed models, algorithms, and software packages can also be used for the safe planning of commercial air transportation. In the near future, tests of a group of agricultural unmanned aerial vehicles are planned.

## 5. Conclusions

An improved algorithm was proposed in this article, using a new strategy for finding a safe route. It is used to effectively solve the problem of planning and analyzing a complex way. In addition, to solve the problem, mathematical modeling, analysis, and evaluation of the complexity of the flight paths of groups of unmanned aerial vehicles, we have proposed a new approach that allows us to compare the complexity of given routes, which is based on splitting ways into parts and analyzing the properties of each of the sections, taking into account their location on the way.

The analysis of the materials for solving the problem showed the feasibility of creating a software application. The main result of using the system is data on the safe flight path for aviation transport systems. In general, the described computational experiment allowed us to establish, using the example of the analysis of the trajectories of unmanned aerial vehicles, the possibility of classifying the laws of the functioning of aviation transport systems by the properties of the spectra. The software application is handy when training specialists in transport security, robotics, and system analysis because it allows you to obtain important information through simulation modeling.

The approach proposed in this article has shown the excellent performance of algorithms. Furthermore, they are more flexible and efficient than most other existing approaches to planning routes of unmanned aerial vehicles.

**Author Contributions:** Data curation, P.N.; methodology, D.S.; writing—original draft, A.K., S.K., E.P. and N.K. All authors have read and agreed to the published version of the manuscript.

**Funding:** This research received no external funding.

**Institutional Review Board Statement:** Not applicable.

**Informed Consent Statement:** Not applicable.

**Data Availability Statement:** Not applicable.

**Conflicts of Interest:** The authors declare no conflict of interest.

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
