# Peer review of "Mathematical Modeling, Analysis and Evaluation of the Complexity of Flight Paths of Groups of Unmanned Aerial Vehicles in Aviation and Transport Systems"

_mathematics, doi:10.3390/math9172171_

Round 1

Reviewer 1 Report

The article does not provide an accurate analysis of the scientific background which is partially explained in the introduction. The Method and the model lack explanations (e.g. what is the formula in line 89? Where does it come from? What are those parameters?). Chapter three how the algorithm builds the routes but it is not clear what the basic hypotheses are and which the thesis. The experimental phase is practically non-existent and it is not clear how the software works.

Author Response

Thank you very much for the reviewer for the comments and suggestions. We have revised the manuscript according to the comments and now explain to the reviewer as follows:

  1. Mathematical modeling of the flight of a group of unmanned aerial vehicles is a key issue in the study of flight safety. A mathematical model for assessing the complexity and safety of the flight path of a group of unmanned aerial vehicles is described, which has shown good results, which allowed us to develop software based on the results of the study that allows us to calculate the complexity and safety of the flight paths of groups of unmanned aerial vehicles, download maps, etc.
  2. When solving the problem, mathematical models known in the theory of regression analysis are used. The mathematical model of air transport was taken as a basis, which is used in many studies, for example, when planning ground operations, as well as when controlling flight processes.
  3. The main hypothesis is that the absolute value of the height difference between the end and start points of any section cannot exceed twice the height difference between the level lines. The routes of unmanned aerial vehicles, in accordance with the task statement, are divided into stages.The values of the property indicators for the sections of each route are calculated using the map. The obtained values are encoded with the symbols of the alphabets A1 and A2.
  4. The software allows you to upload a map, upload data and analyze the route. Laboratory tests were carried out on a small number of drones (up to 7 devices). In the near future, tests are planned for a group of agricultural drones. In the future, it is planned to use the proposed models, algorithms and a software package for the safe planning of commercial air transportation.

Reviewer 2 Report

This article presents a developed software application that is used to improve the efficiency of flight research of groups of unmanned aerial vehicles, based on a new method for assessing flight safety by comparing the complexity of specified air routes. A practical approach to modeling and evaluating the search for a safe way is proposed.

THE WHOLE WORK IS BOTH VERY INTERESTING AND USEFUL.

Points for improvement:

  1. A GOOGLE SCHOLAR literature review made by the reviewer by using the keywords provided by the authors revealed about 15,000,000 works. Please, complete the keywords or/and the literature cited.
  2. It is not clear if time constraints for the flight were considered. Please, make a  comment.
  3. Does the proposed method accounts for multiple tasks? Please, make a relative discussion.
  4. Is there any reward function considered? Please, make a relative discussion. Please, see AUTHOR = {Xie, Songyan and Zhang, An and Bi, Wenhao and Tang, Yongchuan},
    TITLE = {Multi-UAV Mission Allocation under Constraint},
    JOURNAL = {Applied Sciences},
    VOLUME = {9},
    YEAR = {2019},
    NUMBER = {11},
    ARTICLE-NUMBER = {2184},
    URL = {https://www.mdpi.com/2076-3417/9/11/2184},
    ISSN = {2076-3417}
  5. What are the main problems for practical applications?
  6. Does the proposed method accounts for 3D reconaisance?

In my opinion this work could be published after revision.

Author Response

Thank you very much for the reviewer for the comments and suggestions. We have revised the manuscript according to the comments and now explain to the reviewer as follows:

1. A GOOGLE SCHOLAR literature review made by the reviewer by using the keywords provided by the authors revealed about 15,000,000 works. Please, complete the keywords or/and the literature cited.

Reply 1: Time is one of the controlled parameters.

2. It is not clear if time constraints for the flight were considered. Please, make a comment.

Reply 2: First of all, the group control of drones was considered.

3. Does the proposed method accounts for multiple tasks? Please, make a relative discussion.

Reply 3: Yes, we consider the reward function proposed in the work [Se S. et al. Distribution of missions for several unmanned aerial vehicles under restrictions //Applied sciences. - 2019. - Vol. 9. - No. 11 – - p. 2184.]. The reward function takes into account not only the difference in the importance of the goal, but also the limitations of the time window. In addition, an indicator of reconnaissance remuneration is added to the proposed model. The exploration reward is the amount of reward received for exploration in various target areas. It measures the amount of information received and the amount of strategic value.

4. Is there any reward function considered? Please, make a relative discussion. Please, see AUTHOR = {Xie, Songyan and Zhang, An and Bi, Wenhao and Tang, Yongchuan},

TITLE = {Multi-UAV Mission Allocation under Constraint},

JOURNAL = {Applied Sciences},

VOLUME = {9},

YEAR = {2019},

NUMBER = {11},

ARTICLE-NUMBER = {2184},

URL = {https://www.mdpi.com/2076-3417/9/11/2184},

ISSN = {2076-3417}

Reply 4: During the tests, we encountered problems related to the synchronization of software with drones, however, the problem was solved.Also, one of the problems is the high cost of a full-scale experiment, so in this regard, the most effective method of research is mathematical modeling.

5. What are the main problems for practical applications?

Reply 5: This implementation of mathematical models is possible for 3D, but for clarity we use 2D schemes.

Reviewer 3 Report

An improved algorithm was proposed in this article, using a new strategy for finding a safe route. It is used to effectively solve the problem of planning and analyzing a complex way. In addition, to solve the problem, mathematical modeling, analysis, and evaluation of the complexity of the flight paths of groups of unmanned aerial vehicles, authors have proposed a new approach that allows us to compare the complexity of given routes, which is based on splitting ways into parts and analyzing the properties of each of the sections, taking into account their location on the way. It is always advisable to see how mathematics serves other science, and this is the goal of this article. The article has some novelty but requires more clarifications. In order to improve the paper and make it possible for publication, authors need to take the below comments into consideration:

1- As the authors submit the paper to Mathematics, the math part must be strong and solid, authors can recheck the mathematical part.

2- To show the novelty, a comparison between other methods must be added to the paper.

3- A detailed section on the different options and what each button can do in picture 5 must be added. So some case studies and how the software work must be added. 

4- How the software can be embedded into the fdrones?

5- What is the complexity in terms of the execution time of the proposed algorithm?

6- Limitations and future work must be added.

Author Response

Thank you very much for the reviewer for the comments and suggestions. We have revised the manuscript according to the comments and now explain to the reviewer as follows:

1- As the authors submit the paper to Mathematics, the math part must be strong and solid, authors can recheck the mathematical part.

Reply 1: The developed mathematical models are based on well-known approaches to modeling air transportation, published in authoritative and peer-reviewed publications. The results obtained using the proposed new methods and approaches are compared with the results of the application of known methods. The limits of applicability of the proposed methods and approaches are discussed. All the assumptions made during the work are described in detail with a description of the limits of applicability and the impact on the final result. The reproducibility of the obtained results is ensured by a detailed description of all stages of data processing, indicating the parameters of the processing methods, and the software used. The results of theoretical analysis and numerical modeling are confirmed by the results obtained during experimental studies.

2- To show the novelty, a comparison between other methods must be added to the paper.

Reply 2: Currently, there are three classes of approaches to solving the problem of route planning. The first class is based on graph algorithms .The second class is based on a typical heuristic representative. The third class is based on the algorithms of evolutionary calculations. These approaches have a satisfactory performance. They have several disadvantages, such as local optima and a slow global convergence rate. In our work, we propose to use a range of dynamic characteristics to analyze and build an estimate of the complexity of a route represented as a set of sequences of code signs. In this spectrum, dynamic parameters representing the rules for constructing a sequence using recurrent forms of various orders are systematized.

3- A detailed section on the different options and what each button can do in picture 5 must be added. So some case studies and how the software work must be added. 

Reply 3: We have added the new contents on page 10: "At the first stage, a map for simulation is loaded into the software application. Load data allows you to load the data necessary for building a route. Build routers builds a route and analyzes the complexity and safety of the flight paths of a group of unmanned aerial vehicles."

4- How the software can be embedded into the fdrones?

Reply 4: To synchronize the software with drones, the Robot Operating System application development environment and the Gazebo software simulator are used, which allows you to import code written in C-like programming languages. They are used to test algorithms in a simulation environment and break down as little equipment as possible. Both ROS and Gazebo work on Linux — we used Ubuntu 16.04

5- What is the complexity in terms of the execution time of the proposed algorithm?

Reply 5: The main difficulty is the number of drones 

size

complexity

10 20 30 40 50 60
n 0.00001sec 0.00002sec 0.00003sec 0.00004sec 0.00005sec 0.00005sec
n2 0.0001sec 0.0004sec 0.0009 sec 0.0016sec 0.0025sec 0.0036sec
n3 0.001sec 0.008sec 0.027 sec 0.064 sec 0.125sec 0.216 sec

6- Limitations and future work must be added.

Reply 6. The objective function is to maximize the total reward for UAV exploration, where time and resources are subject to many restrictions. Equation (3) is an objective function that indicates that each target area is maximized by multiplying the reward for exploration after the exploration time by the value of the target area. Where xijk is a decision variable, and ci is a fixed coefficient that measures the strategic value of each target area. R(tj) is the reward for exploration of each target area, determined by the decision variable tt. Equation (4) indicates that the total flight time of each UAV to complete missions is less than the maximum flight time of the UAV. Where tijk is the flight time required for the UAV k to fly from the target area i to the target area j, tj is the reconnaissance time allocated by the UAV k to the target area j, as well as the flight time of the UAV k flying to the target area j. Tf is the maximum flight time of the UAV k. Equation (5) indicates that the total exploration time of each UAV is less than the maximum opening time of the reconnaissance sensor. Where Tr is the maximum opening time of the reconnaissance sensor carried by the UAV to. Equations (6) and (7) guarantee that each target area can be explored by one and only one UAV.

Thank you for your valuable comments.

Round 2

Reviewer 2 Report

IN THE PROOFS PLEASE COMPLETE THE KEYWORDS TO ADRESS POINT 1:

 A GOOGLE SCHOLAR literature review made by the reviewer by using the keywords provided by the authors revealed about 15,000,000 works. Please, complete the keywords or/and the literature cited.

Author Response

Thank you for your valuable comment.
We have finalized the list of references and added keywords.

Reviewer 3 Report

Still, section 4 needs a lot of development otherwise the presented software doesn't make sense. The added details are not compatible with figure 5. the description of the figure must be supported by others figures to provide the full functionality of the software.

Author Response

Thank you for your valuable comment.
We have improved section 4, added a description of each button shown in Figure 5. The analyze button allows you to analyze flight paths by the complexity of the specified air routes, allows you to find and select a safe flight path for a set of unmanned aerial vehicles. Load Data allows you to upload data files (synthetic or obtained from drones) in .csv, .txt formats. We have added an additional picture showing a more detailed description of the application.

Round 3

Reviewer 3 Report

The authors didn't change anything between version 2 and version 3, given the impression that they are not taken the reviewer's comments seriously. Despite that, I gave the authors 2 chances to improve their paper to be up to the standard of Mathematics Journal, the authors didn't show any effort. Again the presented software is unclear and I don't think is working, otherwise many illustrations can be added to show the novelty of the paper. 

Author Response

We appreciate your comments, and they are very important to us. In the previous edit, we may not have understood quite correctly how we should reflect the work of the software. In this regard, we have completely removed section 4 "Software application development". We would like to note that the main message of the article is aimed at developing mathematical models for calculating safe flight paths and an algorithm for constructing the optimal flight path of unmanned aerial vehicles. The software is an addition to the main part, but we do not focus on it, so we agree with your comments and we repeat that we have decided to remove this section. We also want to note that at the moment we have developed only a prototype of software for a visual representation of the work of mathematical models, which can work with synthetic data, or already post-processed data obtained from drones. We described the functionality of each button of the prototype software package and demonstrated its interface. 

At the first stage, a map is loaded into the software application to simulate the flight paths of unmanned aerial vehicles. The prototype of the software can only work with maps on which the markup was previously carried out - the coordinates of the map areas. At this stage of research, we carry out the markup in manual mode. At the next stage, we create a data file in which we specify the coordinates of the start and end points of the flight of unmanned aerial vehicles, which is loaded into the program using the "upload data" buttons. This can be a file in the following formats .txt or csv. The "Plot route" button is used to calculate the safe trajectories of unmanned aerial vehicles. All calculations are based on mathematical models and algorithms described in sections 2 and 3. There is also a function for analyzing the safety of the flight path, which is implemented when the "analysis" button is pressed. We have conditionally divided each route according to the level of flight safety (from a high level of danger to a low level of danger), the calculation of which is based on the mathematical models described in section 2. The hazard criterion is the values of the spectra calculated using the mathematical model proposed by us. For greater clarity, in order not to give numerical values of the calculated spectra, we made a conditional color scale, where the high level of danger of trajectories is marked in red (corresponds to high values for the sum of properties P), the safest trajectories are marked in green (corresponds to low values for the sum of properties P).

As laboratory tests, we used data from a full-scale experiment with a small number of drones (we used 7 devices). We received information about the properties of the unmanned aerial vehicle (the properties are described in detail in section 2), which were recorded on the memory card. Next, we worked with this data. In addition, to check the quality of our models and their compliance with experimental data, we used data on drone flights obtained in the sources [34-41].

At the moment, the prototype does not implement software synchronization with drones. We also indicated this in the article. However, we plan to solve this problem using the Robot Operating System application development environment and the Gazebo software simulator, which allows you to import code written in C-like programming languages. They are used to test algorithms in a simulation environment and to disable as little equipment as possible. Both ROS and Gazebo work on Linux - we plan to use Ubuntu 16.04.

We have ideas for further development of the application and suggestions for possible solutions to the synchronization problem. We don't quite understand what drawings you want to add additionally. However, if in your opinion it is necessary, we will be happy to provide you with the drawings that you need, if you tell us in more detail. Thank you again for your valuable comments and sincerely hope that the changes made have improved our work and it meets the standards of the mathematical journal.

Round 4

Reviewer 3 Report

 I see in the attached file authors are revised according to the provided comments. The manuscript can be now accepted.